# Wound Healing Properties of Natural Products: Mechanisms of Action

**DOI:** 10.3390/molecules28020598

**Published:** 2023-01-06

**Authors:** Marilyn S. Criollo-Mendoza, Laura A. Contreras-Angulo, Nayely Leyva-López, Erick P. Gutiérrez-Grijalva, Luis Alfonso Jiménez-Ortega, J. Basilio Heredia

**Affiliations:** 1Centro de Investigación en Alimentación y Desarrollo, A.C. Carretera a Eldorado Km 5.5, Col. Campo el Diez, Culiacán CP 80110, SI, Mexico; 2Post-Doc. CONACYT-Centro de Investigación en Alimentación y Desarrollo, A.C., Carretera a Eldorado Km 5.5, Col. Campo El Diez, Culiacán CP 80110, SI, Mexico; 3Cátedras CONACYT-Centro de Investigación en Alimentación y Desarrollo, A.C., Carretera a Eldorado Km 5.5, Col. Campo El Diez, Culiacán CP 80110, SI, Mexico

**Keywords:** wound healing, natural products, antioxidant, anti-inflammatory, antimicrobial, phenolic compounds, collagen, cell migration

## Abstract

A wound is the loss of the normal integrity, structure, and functions of the skin due to a physical, chemical, or mechanical agent. Wound repair consists of an orderly and complex process divided into four phases: coagulation, inflammation, proliferation, and remodeling. The potential of natural products in the treatment of wounds has been reported in numerous studies, emphasizing those with antioxidant, anti-inflammatory, and antimicrobial properties, e.g., alkaloids, saponins, terpenes, essential oils, and polyphenols from different plant sources, since these compounds can interact in the various stages of the wound healing process. This review addresses the most current in vitro and in vivo studies on the wound healing potential of natural products, as well as the main mechanisms involved in this activity. We observed sufficient evidence of the activity of these compounds in the treatment of wounds; however, we also found that there is no consensus on the effective concentrations in which the natural products exert this activity. For this reason, it is important to work on establishing optimal treatment doses, as well as an appropriate route of administration. In addition, more research should be carried out to discover the possible side effects and the behavior of natural products in clinical trials.

## 1. Introduction

The skin is a fundamental part of the human body, the largest organ, and the first defense barrier against the environment. This organ maintains homeostasis as a reservoir of nutrients, a defense mechanism, and an injury response [1,2,3]. It is made up of multiple layers of ectodermal tissues that protect the muscles, bones, ligaments, and internal organs of our body. A wound is defined as the loss of the integrity and normal functions of the skin that can be caused by physical, chemical, or mechanical agents [4,5,6]. When the skin structure is compromised, its restoration and the recovery of its functions must be carried out as soon as possible to guarantee the organism’s homeostasis, so that the wound healing process begins almost immediately after the skin injury [7].

Wounds can be classified as acute, which takes between 7 and 10 days to heal, or chronic, which may not heal in this amount of time due to a lazy and disorderly healing process. The fact that a wound becomes chronic has been associated with many factors, including underlying disease, infection, prolonged inflammation, the consumption of medications, and oxidative stress related to aging [4,8]. An acute wound can be defined as a recent wound that has not yet progressed through the sequential stages of healing; they can be full-thickness skin lesions involving only the subcutaneous layer of skin or superficial lesions involving the epidermis or superficial dermis, e.g., surgical incisions; thermal injuries; abrasions and lacerations [9]. Good circulation, immune function, and a moist environment free of dead tissue or foreign matter, are seen in acute wound healing, contributing to the normal function of cells and molecules involved in the repair process [10]. Moreover, chronic wounds do not progress through the normal stages of healing and are not repaired in an orderly and timely manner [11]. The resistance to treatment is frequently observed, and it is necessary to prolong the time for the wound to heal, since repair can be hindered by various factors, which can be environmental, such as trauma, desiccation, and infection. Furthermore, host factors, such as a compromised circulation, respiratory function, or immunodeficiency, as well as the presence of free oxygen radicals in wounds, cause the healing process to be interrupted, that can lead to further tissue damage, delaying the normal follow-up phases of the healing process [12]. Wounds can also be classified according to the location, depth, nature, or cause of the injury [13,14].

The treatment of wounds is of great importance for human health, since the lack of healing and the prolongation of treatment time cause increases in economic costs, as well as social implications for health institutions and doctors, the lives of patients, and their families [14,15]. Therefore, improper wound care can lead to more problems in patients, for example, bleeding, infections, inflammation, a poor healing process, and complications in tissue regeneration [16]. According to the World Health Organization (WHO), it is estimated that almost 80% of the world’s population uses traditional practices as health treatment; of these, 85% use plant remedies [17].

Natural compounds from plant extracts can provide a solution to improving the wound healing process by reducing scar formation. Antimicrobial, antioxidant, and healing plant-derived bioactive compounds stimulate the blood coagulation, fight infection, and speed up wound healing [18,19]. There are numerous reports in the literature on natural compounds with the potential to promote wound healing that could be used as an effective treatment throughout the various phases of the wound healing process [20,21,22,23]. These are considered safe, compared to some synthetic molecules, they are more tolerated, and can be much cheaper than conventional therapies [24]. However, more information is needed from the clinical trials, on the use of natural products, in combination with modern drugs, as well as the development of better delivery mechanisms, which represent a very promising area for drug discovery from natural products [25].

In this review, we analyze the evidence from in vitro and in vivo studies, indicating the efficacy and safety of natural compounds, such as polyphenols, terpenes, alkaloids, saponins, and essential oils, in the treatment of wound healing, to establish a summary of the more relevant and up-to-date information on the potential of natural compounds.

## 2. Wound Healing Process

Although the skin is the most exposed and vulnerable tissue, the damage can be completely restored, depending on the individual’s age and health [26,27]. Following the injury, the wound healing process begins, which is a physiological process that involves different types of cells (platelets, macrophages, fibroblast, and epithelial and endothelial cells), extracellular matrix components, cytokines, growth factors, and regulatory molecules; this process also depends on the extent and mechanism of the injury [28,29,30,31]. The wound healing process is divided into four continuous and coordinated phases: hemostasis, inflammation, proliferation, and remodeling (Figure 1) [27,32].

### 2.1. Hemostasis

The first phase, hemostasis, begins immediately when the damage occurs (within a few minutes), altering the integrity of the tissue; at this point, platelets are activated and signal chemicals to form blood clots (composed of fibronectin, fibrin, vitronectin, and thrombospondin) and prompt platelets to form a plug. Furthermore, growth factors (epidermal growth, platelet-derived growth factor, transforming growth factor-β, and chemokines) are released to initiate the repair process [28,33,34,35]. In this phase, prostaglandin H_2_ is converted into thromboxane A_2_ (TXA2) by the action of the enzyme thromboxane synthase. TXA2 is a powerful platelet activator and vasoconstrictor, in addition to participating in the release of macrophages, neutrophils, and endothelial cells, all necessary during the wound healing process [36].

### 2.2. Inflammatory Phase

Once hemostasis is achieved, the inflammatory phase starts, some immune cells begin to reduce the injury, and an immune barrier is established against microorganisms [35,37]. In this phase, leukocytes, especially neutrophils, are recruited to the site of the injury to eliminate bacteria and foreign debris; neutrophils secrete proinflammatory cytokines (promote the expression of adhesion molecules) as an inflammatory response [38,39]. At the same time, monocytes migrate into the wound site and differentiate into macrophages; these cells can recruit additional monocytes and increase their response [34,40]. Neutrophils, monocytes, and macrophages play an important role in this phase due to the release of soluble mediators, such as proinflammatory cytokines (interleukins IL-1, IL-6, IL-8 and tumor necrosis factor α (TNF-α)), and growth factors (platelets derived growth factor (PDGF), transforming growth alpha (TGF-α), transforming growth factor beta (TGF-β), fibroblast growth factor (FGF), insulin-like growth factor-1(IGF-1)) involved in the activation of fibroblasts and epithelial cells needed in the next phase [31].

### 2.3. Proliferative Phase

The proliferative phase takes a few days to weeks and aims to reduce the lesioned tissue area. This phase is characterized by the epidermal regeneration, angiogenesis, granulation tissue formation, and collagen deposition [39,41]. First, the activated fibroblast migrates to the wound and creates extracellular matrix proteins (hyaluronan, fibronectin, and proteoglycan) to replace fibrin clots; then, collagen is synthesized to provide strength to the tissue. This connective tissue or granulation tissue (type 3 collagens is predominant) serves as a scaffold for the formation of new extracellular matrix tissue, new to the network of blood vessels (angiogenesis mediated by endothelial cells) [14,42]. At the same time, cytokines generate re-epithelialization; the epithelial cells proliferate (keratinocytes) and migrate to the wound to form a new epidermal barrier [32]. Furthermore, in this phase, fibroblasts are differentiated into myofibroblasts, decreasing the proliferation and increasing the collagen synthesis, causing the wound contraction and reducing the wound size [37].

### 2.4. Remodeling Phase

In the last phase (maturation and repair), which can take up to 2 years, the extracellular matrix is slowly transformed into a mature scar [14,37]. Here, the collagen is reorganized, the collagen type III produced in the extracellular matrix is replaced by collagen type I (this conversion happens with the closure of the wound), and there is the diminishing of new blood vessels and blood flow; likewise, a cellular environment and mature avascular tissue are formed; finally, the skin only recovers a maximum of 80% of its tensile strength [40,43].

The wound healing process described above occurs in a normal or acute wound, while when there is dysregulation of the immune system during the wound healing process, a chronic wound occurs; that is, there is no orderly repair process, and it can last for a longer period (more than three months), and is not carried out fully (Leong 2016). In chronic wounds, infections and microbial films are persistent; this wound is associated with aging, venous ulcers, high blood pressure, obesity, and diabetes [44,45].

During the chronic wound (Figure 2), the inflammation phase is altered, increasing neutrophils (release metalloproteinases, MMPs), which causes an excess of reactive oxygen species (peroxide anion, hydroxyl ion, and superoxide anion) and destructive enzymes (proteases and elastases). In the same way, inflammatory cytokines are increased (this is due to the excessive increase of proinflammatory macrophages), the availability of growth factors is reduced, and the extracellular matrix components (collagen, fibronectin, and vitronectin) are destroyed, which perpetuates this phase. Similarly, there is impaired angiogenesis, and re-epithelialization is difficult; however, the keratinocytes are hyperproliferative [44,46,47,48,49].

## 3. In Vitro Studies on the Effect of Natural Products on Wound Healing

Several studies have been conducted to evaluate the potential of natural products as a source of compounds or extracts with wound healing properties. However, a prior stage to in vivo studies in the assessment and discovery of novel and natural wound care products is to evaluate their possible effects through in vitro experiments. For instance, keratinocytes and fibroblast cell lines are suitable models to evaluate the wound healing properties of new natural and therapeutic agents [50,51].

According to recent research, a *Jatropha curcas* latex extract spray formulation exerted wound healing properties. Tinpun et al. [52] demonstrated that BJ human fibroblast cells exposed to the *J. curcas* formulation, increased the total soluble collagen (type I) production (47–198 pg/mL) in a dose-dependent manner (7.75–124 µg/mL^−1^ of extract). Furthermore, 24 h of treatment with the *J. curcas* formulation (31 µg/mL) fully exerted the wound healing ability in human keratinocyte (HaCat) and human fibroblast (BJ) cells. The main component in the formulation was the cyclic octapeptide curcacycline A [53]. Additionally, the *J. curcas* latex extract formulation showed an antioxidant effect in the DPPH assay and antibacterial properties against *Staphylococcus aureus*, *Staphylococcus epidermis*, *Escherichia coli,* and *Pseudomonas aeruginosa*, which are required features in products for potential applications in wound healing.

Alilou et al. [54] evaluated the wound repair potential of labdane-type diterpene compounds isolated from the aerial parts of *Rydingia persica*. In this study, LPS (5 µg/mL) was used to downregulate the expression of occludin and claudin-1 (tight-junction proteins involved in the wound healing process) in HaCat cells, therefore reducing the wound-repairing ability of keratinocytes. When HaCat cells were treated with isolated diterpenes (persicunin A, B, D, E, F, H, and calyonol) at 50 µM, the expression of occludin and claudin-1 was significantly increased in the LPS-challenged HaCat cells. Furthermore, it was demonstrated that the diterpenes from *Rydingia persica* exhibited the anti-inflammatory potential by inhibiting the secretion of IL-6 and TNF-α. The authors declare that the evaluated diterpenes might be useful as anti-inflammatory and wound healing agents.

Polysaccharide extracts from algae have also been evaluated to determine their wound healing properties. For instance, Tseng et al. [55] demonstrated that a polysaccharide extract from *Nostoc commune* significantly increased the production of type I collagen (from 0.59 to 1.08 ng/mL) in a concentration-dependent manner (0–5 mg/mL) by a human skin fibroblast cell line (CCD-966SK). Moreover, the extract at 0.0625–1.0 mg/mL significantly reduced the levels of IL-6 in a LPS-challenged RAW 264.7 murine macrophage cell line. The extract contained polysaccharides (221.69 mg/g), sulfates (156.11 mg/g), flavonoids (2.93 mg/g), and phenolic compounds (0.37 mg/g). These results suggest that the polysaccharide extract of *Nostoc commune* exerts effects as a wound healing and anti-inflammatory agent with the potential for application in the cosmetic industry [55].

Kamarazaman et al. [56] evaluated the wound healing effect of the ethanolic extract obtained from *Baeckea frutescens* leaves. In this study, a HaCat and BJ cell line were exposed to different concentrations of an ethanolic extract from *B. frutescens*, and the cell migration rate was determined. The migration rate in HaCat and BJ cells treated with 25 µg/mL and 10 µg/mL of extract, respectively, was similar to or higher than those treated with allantoin (a renovator of epidermal cells and accelerator of wound healing) at 10 µg/mL. Furthermore, it was reported that the extract was rich in flavonoids (containing derivatives of myricetin and quercetin) and condensed tannins and presented an antioxidant capacity, which the authors mention might be contributing to the cell migration properties, and therefore the wound healing potential of the extract from *B. frutescens*.

Table 1 summarizes some of the most recent in vitro studies focused on examining the wound healing effect of diverse natural products. Most of the components present in the extracts or isolated from natural products, exert the in vitro wound healing effect by promoting the cell migration and reducing the cell gap.

## 4. In Vivo Studies on the Effect of Natural Products on the Wound Healing Process

Natural products, such as alkaloids, saponins, terpenes, essential oils, and polyphenols from different plant sources, have been evaluated for their in vitro wound healing and tissue regeneration properties (Figure 3). However, these effects may vary during the in vivo evaluations, due to the many factors that involve biological systems. In this sense, polyphenols have been of particular interest; information about it is described in Table 2.

For instance, the aqueous extracts from the bark of *Waburgia salutaris* were evaluated by Abdelfattah et al. [70]. Due to the extract’s nature, the sample’s main constituents were catechin glucoside, catechin, 11α-hydroxy muzigadiolide, deacetylugandensolide, mukaadial, pereniporin B, and 11α- hydroxycinnamosmolide. Moreover, the wound healing properties of *W. salutaris* were associated with its potential as an anti-aging agent as it decreased the survival rate of *Caenorhabditis elegans* by decreasing its intracellular ROS, decreased the juglone-induced HSP16 expression (a heat shock protein involved in the aging process), and mediated the nuclear localization of the transcription factor DAF-16, which is involved in age modulation and longevity. Furthermore, the mode of action of the *W. salutaris* extract was also associated with the potential of the extract’s compound’s affinity to collagenase and elastase, enzymes involved in aging by breaking the bonds in collagen and elastin.

Moreover, Adadietal. [71] reported that the aqueous extracts of *Cistus ladanifer* were incorporated into a neutral Vaseline cream at 10% and 5%. In this study, sexually mature male rats were used by the carrageenan-induced paw edema and burn wound models. The main constituents in the *C. ladanifer* aqueous extracts were gallic acid, ellagic acid, rutin, catechin, vanillin, and quercetin. The oral administration of the aqueous extract showed anti-inflammatory properties at a concentration of 500 mg/kg, compared to the control drug. Furthermore, the cream with the extract had a healing effect on the burn wound of rats by significatively decreasing the circumference of the edema paw [71].

Akbari et al. [72] evaluated the wound healing effect of the Iranian species *Astragalus floccosus* Boiss. Using male rats that were applied a herbal ointment at 1.5% of *Astragalus* flavonoid-rich fraction, the animals were wounded, and the treatment was compared to silver sulfadiazine (as control). The main phytochemicals in this plant species extracts were formononetin, kaempferol-3-*O*-glucoside, calycosin-7-*O*-β-_D_-glucoside, and kaempferol-3-*O*-neohesperidoside, some other quercetin glycosides and derivatives were also found. Moreover, after the wound assay was performed in rats, the flavonoid-rich extract showed the second-best wound healing result by day 4 and the highest activity three days later. In addition, the flavonoid-rich treatment showed the minimum wound scar. Furthermore, it is suggested that one of the main mechanisms of action in which the *A. floccosus* extracts improved the wound healing was through the modulation of the inflammatory responses. In this sense, flavonoids have been reported as good anti-inflammatory agents, due to their structural characteristics by the modulation of inflammatory cytokines [73].

El Sayed et al. [74] study the phytochemical composition and wound healing properties of leaf extracts from several species of the *Aloe* genus: *A. vera* (L.) Burm. f., *A. arborescens* Mill., *A. eruk*, *A. Berger*, *A. grandidentata* Salm-Dyck, *A. perfoliata* L., *A. brevifolia* Mill., *A. saponaria* Haw. and *A. ferox* Mill. In this study, a chromatographic analysis with HPLC-DAD-MS^2^ showed that the most abundant phenolics were *cis*-*p*-coumaric acid derivatives, 3,4-*O*-(E) caffeoylquinic acid, caffeoylquinic acid hexoside, lucenin II, vicenin II, and orientin. Furthermore, the wound healing activity was assessed by evaluating the anti-inflammatory activity of the methanolic leaf extracts of the *Aloe* species by the oral administration at 250 mg/kg body weight. The wound healing activity is reported using chemically-induced type II diabetic foot ulcer animal models, to which a wound (2 mm × 5 mm skin in full thickness removed) was made on the dorsal surface of the right hind foot of the animals, and then the *Aloe* treatment was topically applied. The treatments resulted in the decreased edema thickness, indicating the aloe extracts’ anti-inflammatory activity. Moreover, the *Aloe* extracts also accelerated the wound healing in diabetic animals. In particular, *A. eruk* showed the best results with a decreased wound area of 86.2% by day 10 of the experiment, hypothesizing that the glycoside aloenin and glucomannans might be the responsible compounds for this effect.

Polyphenols have also been used in biomaterials to evaluate their wound healing properties. For example, Baldwin et al. [75] applied tannic acid incorporated from 0.1% to 1% in collagen beads. The tannic acid-collagen beads have a wound healing effect, due to the antibacterial properties of tannic acid and might be of potential implementation in filling surgical voids at 16 weeks.

Some flavonoids with antioxidant and anti-inflammatory properties have also been evaluated for their wound healing properties. For instance, at concentrations of 0.5, 1, 5, and 50 mM, quercetin was topically applied to wild-type C57BL/6J mice and New Zealand white rabbits under lamellar keratectomy surgery, to evaluate its effect on the corneal scar development, by McKay et al. [76]. The authors report that quercetin at 5 mM reduced the corneal formation in mice by decreasing the size and density of the corneal haze. Furthermore, quercetin at 50 mM fell by 2.2-fold, the scar severity by week 1, and 5 mM quercetin lowered the scar severity by week 2. Moreover, they found that 5 mM quercetin treatment reduced the stromal backscattering in rabbits. The authors suggest that quercetin’s possible mechanism of action might be attributed to the ability to modulate the transforming growth factor β (TGF-β).

Peppermint essential oil has shown wound healing properties, as reported by Ghodrati et al. [77]. The authors used commercial peppermint essential oil (PEO-NLC) loaded in lipid nanocarriers in this study. The encapsulated and non-encapsulated peppermint essential oil were then applied to mice subjected to two circular full-thickness skin wounds at the dorsal inter-scapular region; after wounding, the animals were treated with 25 × 10^7^ cells of the *P. aeruginosa* and *S. aureus* strains. The PEO-NLC accelerated the wound healing which they associated with the antimicrobial properties of the constituents of the peppermint essential oil; this antimicrobial activity could also be related to reducing the wound inflammation and thus improved the healing activity. The main components of the PEO-NLC were menthol (39.80%) and mentone (19.95%).

Another study by Alam et al. [78] used clove oil-loaded nanoemulsions that were orally administered to female albino Wistar rats, subjected to excision wounding. Pure and encapsulated clove essential oil were administered at 25 mg/kg for 10 d. They showed that the wound contraction in the animals increased until day 24 in pure and the clove essential oil-loaded nanoemulsion, showing a similar effect to that of the antibiotic gentamycin.

**Table 2 molecules-28-00598-t002:** In vivo studies of the wound healing properties of the natural products.

Source	Main Components	Model	Treatment	Key Results	Type of Lesion	Reference
*Fraxinus* *angustifolia*	Rutin, quercetin, tannic acid, catechin	Male CD-1 mice (5–6 weeks old, 25–35 g)	Cutaneous inflammation was induced by phorbol ester. Polyphenol extract was incorporated into liposome-like vesicles and applied	Leaf-loaded EG-PEVs reduced edema size and myeloperoxidase activity	Chronic	[79]
*Bletilla striata*	Protocatechuic acid, p-hydroxybenzoic acid, caffeic acid, p-hydroxy- benzaldehyde, 3-hydroxycinnamic acid, and ferulic acid	Outbred male mice (3–8 weeks old, 18–22 g)	One mL of *Bletilla* ointment was applied to burn mice 2 times per day for 15 days	The *Bletilla* ointment-treated mice improved the wound healing rapidly by day 5 and almost fully healed by day 13	Acute	[80]
*Artemisia judaica*	Piperitone, terpinene-4-ol, α-thujone, β-thujone, 1,8-cineole, camphor, linalool	A skin burn induction model was used on Sprague Dawley female rats (150 g, 3 months old)	Ointment with 5% *A. judaica* essential oils applied twice per day to rats with second degree burns for 21 days	The *A. judaica* ointment improved the wound healing process by increasing SOD and CAT levels, reduced the pro-inflammatory marker TNF-α and increased the anti-inflammatory TGF-b1 and IL-10 levels	Acute	[81]
*Mentha longifolia* subs. *Typhoides* and *schimperi*	Subsp. typhoides: piperitenone oxide, piperitone oxide. Subsp. schimperi: pulegone, menthone	Wistar albino mice (25–35 g) were subjected to second-degree burn wounds.	Essential oils were incorporated into an ointment and applied daily to mice for 21 days	The *A. judaica* essential oils showed antimicrobial activity against *E. coli*, *K. pneumonide*, *S. aureus*, *B. cereus*, and *C. albicans.* The subsp. typhoides showed better wound healing activity with skin regeneration, new skin appendages formation, and increased deposition of collagen and number of fibroblasts	Acute	[82]
*Thymus vulgaris* honey	Thymus Essential oils from *O. vulgare*, *Rosmarinus officinalis*, and *Thymus vulgaris*	Wistar rats (180–200 g, 3–5 months) were induced to acute dermal toxicity. Rabbits were used for wound healing test	Acute dermal irritation test: essential oils from each plant species were used at 0.5 and 5%. Wound healing assay: treatments with a mix of *T. vulgaris* honey and *O. vulgare* essential oil, mix of *T. vulgaris* honey and T. *vulgaris* essential oil, and a mix of *T. vulgaris* honey and *R. officinalis* essential oil were applied every 24 h for 5 days	The highest wound healing activity was shown for the mix of *T. vulgaris* honey and *T. vulgaris* essential oil, with wound closure rates for both thermal and chemical-induced burns of 85.21% and 82.14%, respectively	Acute	[83]
*Rosmarinus* *officinalis*	Essential oils of *R. officinalis* were incorporated into nanoencapsulated lipid nanocarriers	Mice	The animals were treated daily for 14 days with a gel containing 0.5% essential oil and a 0.5% gel with nanoencapsulated essential oils	Both *R. officinalis* essential oil treatments showed lower wound areas, exerted antimicrobial activity, increased levels of the anti-inflammatory cytokines IL-3, IL-10, VEGF, and SDF-1α	Acute	[84]

Our literature review found no consensus on the effective concentrations at which natural products exert their wound healing activity. In this sense, two main aspects of the biological properties of natural compounds focused on during the wound healing assays, are their antimicrobial and anti-inflammatory properties. Both essential oils and polyphenols have been associated with these properties, related to their chemical structure, bioavailability, and administration route [73,85,86,87]. Moreover, natural products, such as essential oils and polyphenols, are produced in plants by the secondary metabolism aiming to protect the organisms against biotic and abiotic stresses. Thus, the content and distribution of these molecules in plants can change depending on many factors [88]. As a result, the formulations derived from plants can vary their phytochemical content, which will intrinsically affect their bioavailability and bioactivity. For this reason, we suggest that either: (a) the main compounds responsible for the wound healing effects are identified in each plant species, and the formulations are designed using these molecules, and (b) if there is a need to use plant-based products, controlled crops are developed to minimize the effect of biotic and abiotic stress on the phytochemical content of the plant species.

## 5. Mechanisms of Action of the Natural Products in Wound Healing

The main mechanisms involved in the potential of natural products in wound healing treatments are antioxidant, anti-inflammatory, and antimicrobial activities. In many works, both in vitro and in vivo, the results of plant extracts that contain natural compounds that have more than two of these characteristics are shown, which makes them the target of future research. The importance of these characteristics during the wound healing process is briefly described below, as well as the relationship between them.

### 5.1. Antioxidant

Reactive oxygen species (ROS) are small molecules derived from oxygen that act as oxidizing agents and contribute to cell damage; therefore, they play a crucial role in preparing the normal wound healing response [89]. ROS can protect tissues against infection at low levels and stimulate effective wound healing, by stimulating cell survival signals [90,91]. However, oxidative stress is generated when an excess induces cell damage and a proinflammatory state [92]. Antioxidants, such as polyphenols, are chemical compounds that can donate their electrons to other molecules, such as ROS, thus avoiding the sequestration of electrons to other molecules of biological importance, such as some proteins or DNA. In addition, antioxidants activate a complex cascade of reactions to transform ROS into more stable molecules; thus, these substances maintain non-toxic levels of ROS in wound tissues, contributing to enhancing the healing process [93,94].

### 5.2. Anti-Inflammatory

The inflammatory phase is an essential part of the wound healing process; however, excess inflammation and its prolongation during the initial stages of this process can cause scarring, fibrosis, and a delay in the healing process [95,96]. In addition, inflammation is important to prevent microbial infection and remove dead cells and cellular debris. During this stage, proinflammatory mediators, such as interleukins, TNF-α, inducible nitric oxide synthase, and chemokines are generated [97,98]. Macrophages with the proinflammatory M1 phenotype are the first responders during the repair stages. In contrast, those with the pro-repair, anti-inflammatory M2 phenotype are found in greater numbers during the later stages of wound healing [99].

### 5.3. Antimicrobial

Skin wound infections can be differentiated into minor superficial, or life-threatening infections, according to their etiology and the severity of the microbial invasion [100]. At the beginning of the infectious process, the organisms with the greatest presence in the wound are gram-positive, such as *Staphylococcus aureus* and *Streptococcus pyogenes*, while gram-negative organisms, such as *Escherichia coli* and *Pseudomonas aeruginosa,* appear in later stages of the process, and are associated with the development of a chronic wound [101]. In the normal process of wound repair, infection is prevented by the immune system’s rapid response to eliminate the invading pathogens. In general, macrophages participate in this process, migrating to the wound site and subsequently engulfing the bacteria; there is also the activity of the helper T cells, which secrete interferon-γ and the CD40 ligand, and coordinate the adaptive and humoral immune response for the elimination of invading pathogens [102]. When the immune system is unable to eliminate the pathogen, infection results, causing the deterioration of the granulation tissue, growth factors, and extracellular matrix components, disrupting the normal healing process [103,104,105].

Extracts rich in phenolic compounds from *Diospyros mespiliformis*, *Haplophyllum tuberculatum,* and *Mangifera indica* showed important antioxidant, anti-inflammatory, and antimicrobial activities when evaluated in different models of wound healing [106,107,108]. Rocha et al. [109] evaluated the antioxidant, anti-inflammatory, and healing activities of *Artemisia campestris* L. hydrodistillation residual water and its essential oil, finding that the residual hydrodistillation water showed promising antioxidant and anti-inflammatory activities without causing toxicity in the macrophages and fibroblasts. The essential oil showed an antioxidant effect and antifungal activity, especially against *Epidermophyton floccosum* (causing superficial and cutaneous mycoses) and healing activity. Moreover, the *Zizyphus mauritiana* fruit extract suppressed the hydrogen peroxide (H_2_O_2_) radical formation in a dose-dependent manner, demonstrating consistent antioxidant activity with IC_50_ of 189.2 μg/mL^−1^. In this work, the antioxidant activity of the extract was attributed to its superoxide dismutase (SOD) activity and H_2_O_2_ scavenging ability, which can remove ROS and improve the wound healing process, which could be attributed to its content of phenolic compounds [110]. The *Bacopa monniera* extract at 25 mg/kg of body weight, increased the levels of SOD, glutathione, and catalase in the granulation tissue of the group subjected to treatment for 20 d; this antioxidant activity was associated with the significant reduction of the area of the scar (75.2 ± 4.04 mm^2^), compared to the control group (99.8 ± 4.92 mm^2^) [111]. The topical treatment with 10% *Schinus terebinthifolius* oil ointment reduced the accumulation of leukocytes and proinflammatory mediators, such as IL and TNFα. A significant reduction was observed in TNF-α, CXCL-1, and CCL-2 levels and in the accumulation of neutrophils and macrophages [112]. The *Croton bonplandianum* latex extract and α-tocopherol in wounded rats demonstrated a greater antioxidant activity, a significant rate of wound contraction, and a higher collagen content than other herbal treatments. In addition, the increased concentrations of SOD and catalase were found, as well as reduced concentrations of lipid peroxidation products in the granulation tissue [113].

Natural products from rosehip oil promoted wound healing by facilitating the transition of the M1 to M2 macrophage phenotype, increased the collagen III content in wound tissue, and inhibited epithelial-mesenchymal transition during wound healing to improve scarring [114]. Essential oils, according to some authors, have antimicrobial activities in wound healing, and this is attributed to their content of phenolic compounds, mainly thymol and carvacrol [105,115]. The possible mechanism of action is that the natural compounds in essential oils attack the phospholipids present in the cell membranes and the lipids available in the bacterial cell wall, causing an increase in the permeability and later cell lysis [116]. Extracts of *Angelica dahurica*, *Rheum officinale,* and *Euterpe oleracea* demonstrated reduced plasma levels of TNF-α, TGF-β1, IL-6, and IL-1β, compared to the treatment groups, favoring the inflammation phase of the wound healing process [117,118].

## 6. Challenges and Solutions in the Use of Natural Products, as Wound Healing Agents

One of the qualities of natural products is their versatility in uses, routes of administration, and compatibility with vehicles that release them [119]. The pharmaceutical industry has exploited these benefits by producing multiple cosmetic, dermatological, and medicinal products that help to improve skin health and wound healing [120]. In addition, these products can be incorporated into formulations as bioactive agents [121]. The research and development departments constantly innovate, rediscovering natural, profitable, and non-cytotoxic sources. Once the ideal plant material has been identified, it is recommended that the sole supply be maintained or have its crops standardize management and enhance the production of the metabolites using genetic engineering tools; design efficient and intensive extraction processes [122]. Figure 4 shows the unit operations for extracting and using natural products in the pharmaceutical industry.

Due to their chemical nature, natural products can present certain challenges in their handling and incorporation into pharmaceutical products, such as creams, lotions, patches, and gels, among others. One of the great challenges is their low extraction yield since they are found in small proportions [123], so their industrial use is limited by the high volumes necessary to produce them in series.

Another of the great challenges that researchers, is that to extract the metabolites from plant tissues, it is necessary to use solvents, whether lipophilic or hydrophilic, or of medium polarity (acetone, dichloromethane, ethyl acetate). However, using toxic solvents, such as hexane, methanol, and dichloromethane, limits their applications. Even though they are removed by vacuum-assisted concentration or lyophilization, traces of them can be found, causing irritation and cytotoxicity. Furthermore, if the exposure is chronic, it promotes the development of chronic degenerative pathologies [124,125].

Some of these metabolites, such as flavonoids and terpenes, are thermolabile. During the production process, can be degraded or undergo biotransformation, exerting an unwanted effect or simply not reaching their target. Therefore, these metabolites must be added during non-thermal operations that allow their easy incorporation and stability. Likewise, they use inert excipients that do not interact with the product; the packaging must protect against UV light, humidity, and foreign matter for the compounds to reach the end user in the ideal concentration [126,127].

Other challenges in the use of natural products, particularly alkaloids [128,129] and terpenes [130], is their toxicity, in addition to the fact that they can cause irritation or allergic reactions [131], so the doses must be well administered and provide moisturizing vehicles and be capable of protecting the compound and the skin, mainly.

One of the potential solutions to these challenges is green solvents, such as supercritical fluids [132] and deep eutectic solvents [133], which are emerging technologies capable of extracting large amounts of bioactive compounds. In addition, they are biodegradable, economical, innocuous, and highly versatile since they can extract both hydrophilic and hydrophobic compounds [134,135]. Likewise, these have a great advantage: they can be applied directly or added to the final product without the need to remove the solvent. Hence, production costs are lower, saving energy and material resources [136].

To stabilize and protect the metabolites, a series of biomaterials capable of prolonging their useful life, has been designed and developed, functioning as releasers at pH changes or acting as sensors [137,138]. Furthermore, these biomaterials (microcapsules, nanocapsules, gels, hydrogels) can exert bioactive properties by themselves, so in addition to exerting properties as a vehicle for these compounds, they can contribute to healing [139,140,141].

## 7. Perspectives and Conclusions

Natural products and secondary metabolites from different plant sources have been used to treat wounds for centuries, worldwide. The most common mechanisms behind the phytochemical-mediated optimization of the wound healing process are their antioxidant, anti-inflammatory, and antimicrobial activities. In this review, we can find the most current reports, both in vitro and in vivo, which provide evidence to support the premise that natural products, belonging to different groups of phytochemicals, exert a potential for wound healing and the underlying molecular mechanisms; These compounds can be involved in the four stages of the wound healing process, since they interact at the intracellular level, for example, in the modulation of the generation of ROS, increasing the response of immune cells to decrease the inflammation and regeneration of the tissue, while reducing the probability of infection by helping to eliminate invading pathogens from the wound. Furthermore, natural compounds from plant sources contribute to the normal development of the healing process, avoiding the deterioration of the granulation tissue, and helping the proper functioning of the growth factors and components of the extracellular matrix. Moreover, an important aspect to consider is that the content and distribution of bioactive compounds in the different plant extracts may be affected by biotic or abiotic factors to which the source of the studied compounds has been exposed. Therefore, plant skin formulations can vary their phytochemical content, which will intrinsically affect their bioavailability and bioactivity.

For this reason, we believe it is important to identify the main compounds responsible for the healing effects in each plant species and, from this, design formulations using these pure molecules. In addition, carrying out controlled cultures of the species of interest could minimize the effect of biotic and abiotic stress on the phytochemical content of plant species; It is important to also work on establishing optimal doses, as well as an adequate administration route for these treatments. Finally, the use of natural compounds for the treatment of wounds requires more research, also to know their side effects and their behavior in clinical trials.

## Figures and Tables

**Figure 1 molecules-28-00598-f001:**
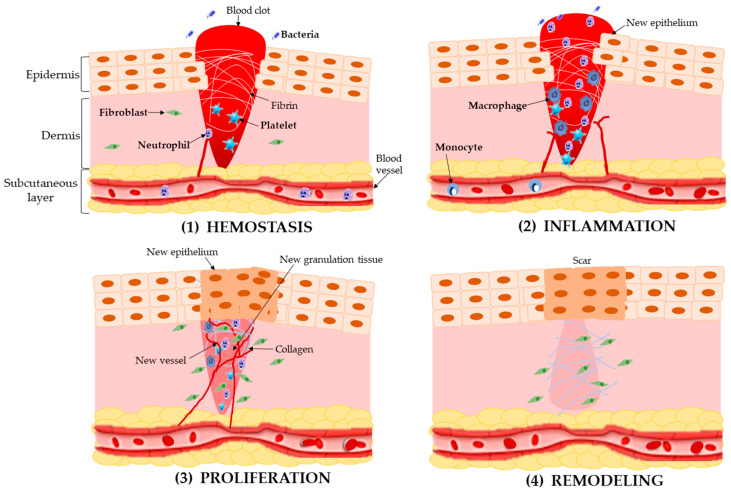
Wound healing process. (**1**) Hemostasis (coagulation); (**2**) inflammation (early/late inflammation); (**3**) proliferation (proliferation/migration/epithelialization/granulation); (**4**) remodeling (maturation/repair).

**Figure 2 molecules-28-00598-f002:**
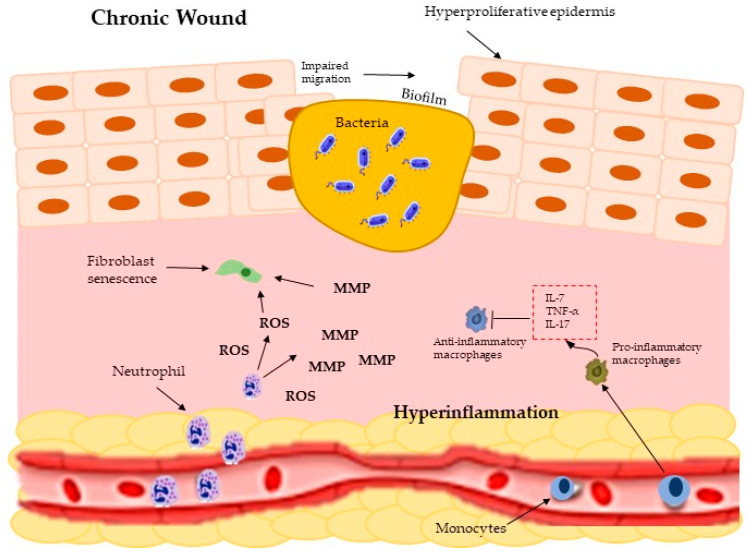
Chronic wound exhibit and persistent inflammation, an infection (biofilm), hyperproliferative epidermis, fibroblast senescence, elevated MMPs (metalloproteinases), impaired migration.

**Figure 3 molecules-28-00598-f003:**
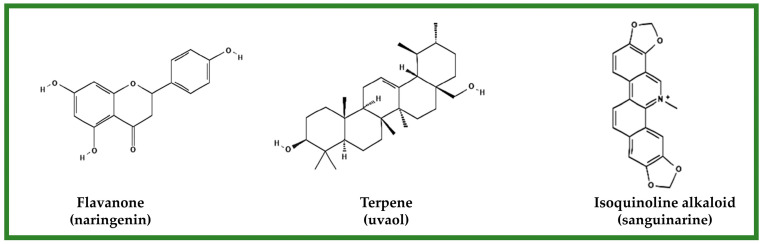
Example of the compounds from natural products evaluated for their in vitro wound healing effect.

**Figure 4 molecules-28-00598-f004:**
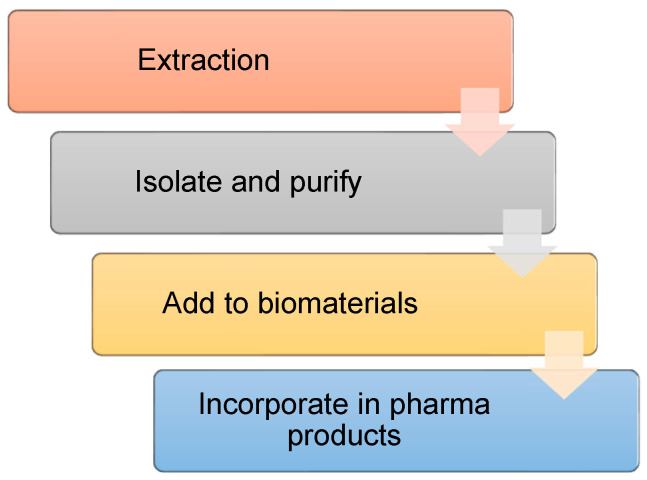
Operations for obtaining and incorporating secondary metabolites in dermatological products. During the extraction, it is necessary to use green and innocuous solvents to extract large amounts of the compounds of interest. During its isolation and purification, column and thin-layer chromatography are recommended, as well as spectroscopic tools, to elucidate and monitor the compound of interest. Once the metabolite is isolated and purified, it is necessary to use biomaterials that protect it and, at the same time, be innocuous and biodegradable. It is worth mentioning that before their incorporation into pharmaceutical products, it is necessary to evaluate the biological, functional, and in vitro and in vivo tests. It is worth mentioning that the incorporation into the final product must be during operations that safeguard its bioactivity and stability.

**Table 1 molecules-28-00598-t001:** In vitro studies evaluating the wound healing potential of natural products.

Source (Type of Extract)	Main Components	Cell Line/Dose *	Key Results	Reference
*Aristolochia saccata* leaves (methanolic extract)	NE	Mouse fibroblasts (L929)/250 µg/mL, 48 h	↑Cell migration ↑Collagen-1 expression	[57]
*Bahuinia ungulate* stem wood (ethyl acetate fraction)	Phenolic compounds, tannins, fisetinidol (flavanol)	Human epithelial (A549)/10 and 100 µL, 24 h	↑Cell migration	[58]
*Entada phaseoloides* leaves (acetone extract)	Protocatechuic acid, epicatechin, kaempferol	Human keratinocyte (HaCat)/100 µM	↑Cell migration during wound closure	[59]
*Fumaria parviflora* aerial parts (ethanolic extract)	Sanguinarine (isoquinoline alkaloid)	Human skin fibroblasts/1 µg/mL, 24 h	↑Cell migration ↑Wound closure percentage	[60]
*Hypericum pseudolaeve* aerial parts (methanolic and aqueous extracts)	Gallic acid, catechin, chlorogenic acid, syringic acid, epicatechin, rutin, quercitrin, and quercetin, among other phenolic compounds	L929/62 µg/mL, 24 h	↑Wound closure percentage	[61]
*luchea indica* branches (ethanolic extract)	Flavonoids, phenolic compounds, tannins, alkaloids, and terpenoids	Primary epidermal keratinocytes/62.5 µg/mL, 2 h Human dermal fibroblasts/62.5 and 125 µg/mL, 2 h Oral mucosal keratinocytes (HO-1-N-1)/62.5 and 125 µg/mL, 2 h	↓Cell gap area	[62]
*Rhodomyrtus tormentosa* leaves (ethanolic extract)	Saponins, flavonoids, steroids, tannins, phenolic compounds	Human fibroblasts/62.5 µg/mL, 24 h	↑Rate of cell migration	[63]
*Parrotia persica*	Myricetin-3-*O*-β-rhamnoside, chlorogenic acid	Normal human keratinocyte (NHEK), normal human dermal fibroblast (NHDF)/ 10 µg/mL, 24 h Human umbilical vein endothelial (HUVEC)/10 µg/mL, 8 h	↑Wound closure percentage (NHEK, NHDF) ↑Number of tubular network formation (HUVEC)	[64]
*Senna auriculata* leaves (methanolic extract)	Mome inositol, 13-docosenamide, (Z)-, cycloheptasiloxane, tetradecamethyl-, and octadecanoic acid, 2-hydroxy-1- (hydroxymethyl)ethyl ester, alkaloids, flavonoids, phenolic compounds, and tannins	L929/25 and 50 µg/mL, 24 h	↑Rate of wound healing	[65]
*Sorocea guilleminiana* leaves (infusion)	Salicylic acid, gallic acid, pinocembrin, and isoquercitrin, among other phenolic compounds	Mouse embryo fibroblast (NIH-3T3)/ 4 µg/mL, 18 h	↑Cell migration/proliferation	[66]
Standard	Naringenin (flavanone)	HaCat/200 µM, 24 h	↑MMP-2, MMP-9, MMP-14, VEGF-A expression and ↑Cell migration	[67]
Standard	Uvaol (terpene)	NIH-3T3 and endothelioma tEnd.1/50 µM, 24 h	↑Cell motility and migration ↑Fibronectin and laminin deposition in fibroblasts	[68]
Standard	Vitexin (glycosyl-flavone)	NIH-3T3 and HaCat/1 µg/well, 72 h	↑Cell migration	[69]

* Concentrations reported did not show cytotoxic effects; MMP: matrix metalloproteinase; NE: not evaluated; VEGF: vascular endothelial growth factor; ↑: Increased; ↓Decreased.

## Data Availability

Not applicable.

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
