# Peer review of "Wound Healing Properties of Natural Products: Mechanisms of Action"

_molecules, 2023, doi:10.3390/molecules28020598_

Round 1

Reviewer 1 Report

The authors have summarized various natural products that have an impact on the wound-healing process. The article includes various aspects and stages of wound healing and how natural products might take part in this process. However, there are a few points that need to be addressed;

1.       The authors need to discuss the clinical relevance of these or some of these natural products. Is there clinical evidence for these natural products?

2.       It would be interesting to know if natural products have been used in combination with other drugs such as antimicrobials. The authors can discuss their opinion on how natural products should be used in wound healing.

3.       Authors need to discuss the reports on the side effects of these natural products from in-vivo studies.

4.       The authors need to indicate which natural products are studied for acute and chronic infections. The authors have mentioned chronic and acute wound infections in general but there is no information about them. It would be good to add a column in the table indicating whether they have been used for chronic or acute infections.

5.       How to increase the clinical efficacy of these natural products?

Author Response

File has been added.

Reviewer 2 Report

The manuscript study "Wound-healing Properties of Natural Products: Mechanisms of Action". The idea and information provided are interesting. This article requires some improvements before publication. Below are some of the technical and non-technical points which should be addressed in order to move on:

It is suggested to mention the challenges of using these compounds. The problems of direct use of these compounds or their instability should be stated. Solutions are also provided.

It is suggested that the innovation of this research be highlighted.

It is suggested to express the potential of industrial applications of Natural Products.

Conclusions should be rewritten based on key research findings.

Author Response

File has been added.

Reviewer 3 Report

The manuscript describes about ‘Wound-healing Properties of Natural Products: Mechanisms of Action’. The manuscript is well organized and based on solid scientific data. I strongly recommend this manuscript for publication with minor revision. My specific comments are as below,

Comment 1 (line 16) - Authors are asked to elaborate on various types of wounds and their associated wound healing processes. 

Comment 2: The majority of the wound healing studies included in this manuscript are extract-based. I would like the author to include more examples of isolated pure compounds along with their structures for the readers' benefit. Furthermore, only polyphenolic compounds are discussed, with no information on other classes of secondary metabolites such as alkaloids, terpenoids, and so on. 

Comment 3 - Many traditional plants are also used to treat wounds (e.g. Curcuma longa, Tridax Procumben etc). The review is lacking in information on traditional plants used in wound healing. 

Comment 4: It is pointless to number section 4.1 for polyphenolic compounds because it is only a subheading in section 4.0. 

Overall, there is a lot more room for writing about isolated pure natural compounds and traditional plant species involved in wound healing. I believe the author should ensure that the most common examples of wound healing medicinal plants are included in the manuscript.

Author Response

File has been added.

Round 2

Reviewer 1 Report

Manuscript can be accepted after incorporating all the suggestions by reviewers.